# Phospholipase D1 Attenuation Therapeutics Promotes Resilience against Synaptotoxicity in 12-Month-Old 3xTg-AD Mouse Model of Progressive Neurodegeneration

**DOI:** 10.3390/ijms24043372

**Published:** 2023-02-08

**Authors:** Chandramouli Natarajan, Charles Cook, Karthik Ramaswamy, Balaji Krishnan

**Affiliations:** 1Mitchell Center for Neurodegenerative Diseases, University of Texas Medical Branch, 301 University Blvd., Galveston, TX 77555-1045, USA; 2Graduate School of Biomedical Sciences, University of Texas Medical Branch, 301 University Blvd., Galveston, TX 77555-1045, USA; 3School of Medicine, University of Texas Medical Branch, 301 University Blvd., Galveston, TX 77555-1045, USA; 4Department of Neurology, University of Texas Medical Branch, 301 University Blvd., Galveston, TX 77555-1045, USA

**Keywords:** excitability, neurodegeneration, synaptic homeostasis, cognitive deficit, Alzheimer’s disease, phospholipase D, dyslipidemia

## Abstract

Abrogating synaptotoxicity in age-related neurodegenerative disorders is an extremely promising area of research with significant neurotherapeutic implications in tauopathies including Alzheimer’s disease (AD). Our studies using human clinical samples and mouse models demonstrated that aberrantly elevated phospholipase D1 (PLD1) is associated with amyloid beta (Aβ) and tau-driven synaptic dysfunction and underlying memory deficits. While knocking out the lipolytic *PLD1* gene is not detrimental to survival across species, elevated expression is implicated in cancer, cardiovascular conditions and neuropathologies, leading to the successful development of well-tolerated mammalian PLD isoform-specific small molecule inhibitors. Here, we address the importance of PLD1 attenuation, achieved using repeated 1 mg/kg of VU0155069 (VU01) intraperitoneally every alternate day for a month in 3xTg-AD mice beginning only from ~11 months of age (with greater influence of tau-driven insults) compared to age-matched vehicle (0.9% saline)-injected siblings. A multimodal approach involving behavior, electrophysiology and biochemistry corroborate the impact of this pre-clinical therapeutic intervention. VU01 proved efficacious in preventing in later stage AD-like cognitive decline affecting perirhinal cortex-, hippocampal- and amygdala-dependent behaviors. Glutamate-dependent HFS-LTP and LFS-LTD improved. Dendritic spine morphology showed the preservation of mushroom and filamentous spine characteristics. Differential PLD1 immunofluorescence and co-localization with Aβ were noted.

## 1. Introduction

Alzheimer’s disease (AD) is the most common cause of dementia and the sixth leading cause of death that cannot be prevented, cured or even slowed [1]. In addition, of the 55 million AD patients worldwide, two thirds are women [2,3,4,5,6]. Therefore, there is not only an urgent need to find a cure, but also an emerging need to investigate therapeutics in a gender-specific manner. Recent therapeutic approaches target the molecular mechanisms causing synaptic dysfunction (an early event leading to memory deficits) driven by low molecular weight aggregates of Aβ (AβO) and tau (TauO) [7,8,9,10,11,12,13,14,15].

Robust evidence from earlier studies [16,17,18] reported elevated phospholipase activity in post-mortem AD brains. Clinical trials using phospholipase substrates, phosphatidyl choline (PC) or lecithin, demonstrated a temporary improvement in memory in AD patients [19]. These data suggest that decreasing such activity could be important in the therapeutic attenuation of cognitive decline. Mammalian phosphatidyl-choline phospholipase D (PC-PLD or PLD) is a family of lipolytic enzymes that can affect membrane curvature, exocytosis, endocytosis, vesicle release and neurite outgrowth, all of which are important for maintaining synaptic function [20,21,22,23,24,25,26,27]. While knocking out the gene-encoding PLD is not detrimental to survival across species [28,29,30], the elevated expression of PLD is implicated in cancer, cardiovascular disease and neuropathologies [31,32]. Thus, there has been great interest in developing small molecule inhibitors for mammalian PLD isoforms [33,34,35,36,37,38]. More importantly, upregulated PLD1 expression and activity was reported in reactive astroglial cells from AD brains, where a physical interaction with amyloid precursor protein (APP) was noted [39]. PLD1 activation was also implicated in the mitochondrial dysfunction in the brains of scrapie-infected mice [40] as well as in AD brains [41], thus increasing the importance of downregulating PLD1 levels as a potential therapy. Our study reinforced the human clinical relevance of the involved isoforms in neuronal function [15].

We investigated whether reducing PLD1 levels using a specific small molecule inhibitor in a 3xTg-AD mouse model improved synaptic function underlying AD-like neuropathology [42]. This model contains AD-related human variant genes for amyloid beta precursor protein (APPswe), presenilin 1 (PS1M146V) and microtubule-associated protein tau (tauP301L). The 3xTg-AD is one of the two models widely used to study interventions against human amyloid and tau pathology simultaneously. Recent studies have also outlined sex-specific differences at later ages in 3xTg-AD cognitive responses, making our approach timely and necessary [43,44,45,46,47,48,49,50]. The presence of Aβ plaques were reported in 3-month-old 3xTg-AD mice along with extracellular plaques in the neocortex and hippocampus at six months [51]. Six-month-old 3xTg-AD mice showed decreased hippocampal synaptic physiology with behavioral changes [43] that mirrored the human Aβ-related AD pathology [52]. However, tau expression and underlying pathological changes were prominently observed in the 3xTg-AD mice hippocampal regions only from 9 months onward. Therefore, understanding tau-related effects necessitates the use of mice aged 9 months and above.

We were the first to observe that the inducible isoform, PLD1, but not the constitutive isoform, PLD2, shows elevated synaptosomal expression (thus complementing the earlier studies that established that PLD1 was also aberrantly elevated in astrocytes and mitochondrial fractions) in the post-mortem hippocampi of AD brains compared to age-matched controls. Moreover, our systematic functional studies established that increased PLD1 promotes synaptic vulnerability and can exacerbate oligomeric (AβO- and TauO-) amyloidogenic cognitive decline. Our follow-up study investigated the scientific premise of progressive Aβ accumulation in 6-month-old 3xTg-AD mice expressing elevated PLD1 [42], and more importantly, established PLD1 inhibition as a viable therapeutic target in mediating cognitive decline against Aβ. We were the first to report that prolonged administration of VU 0155069 (also referred to as VU01) was sufficient to preserve dendritic spine integrity and preserve hippocampal synaptic function and prevent underlying memory deficits in 3xTg-AD mice. Since Aβ and tau accumulation continue increasing in 3xTg-AD with age, and our data suggesting crude synaptosomal PLD1 expression also increases from 6 months to 18 months [15], we asked whether PLD1 inhibition was effective at 12 months, when the levels of tau are significantly higher, than at 6 months. Further, at 12 months, 3xTg-AD mice display many senescence markers, including neuroinflammation [53]. By performing systematic studies at two different ages, 6 months (~20–30 years old equivalent for human age) and 12 months (~38–47 years old equivalent for human age), we addressed the scientific premise and rationale for the suitability of using PLD1 inhibition (with VU01) as a therapeutic agent over an appropriate age range in terms of human age equivalence. More importantly, this study is among the few to address whether therapeutic intervention with the same dosage of VU01 after the full-blown effects of Aβ and tau is efficacious, a premise that would be extremely important when advancing to clinical trials to estimate whether this intervention can be applied at early stages or even when it is late stage.

Recent observations put more emphasis on tau-dependent signaling events over Aβ effects due to its predominance in driving memory deficits and neurodegeneration in the later stages of progressive cognitive decline [54,55,56]. Emerging studies also propose that reducing tau can ameliorate Aβ effects and confer significantly greater improvements in the performance of neuronal circuits, regardless of Aβ and/or tau clearance [56,57,58,59,60].

Thus, in this study, we addressed whether chronic PLD1 inhibition in the later stages of AD-like neuropathology, that has a greater dependence on tau, is still sufficient to provide resistance to the progression of cognitive decline. We (1) looked at whether the dosage at 1mg/kg once every 2 days for 30 days was sufficient to reduce PLD1 expression (using immunofluorescence); (2) studied the effect on dendritic spine morphology (using Golgi–Cox staining) as the underlying mechanism of action that is preserved by inhibiting PLD1; (3) observed for improved hippocampal synaptic function (excitatory and inhibitory) in the Schaffer collateral synapses (electrophysiology); and (4) investigated the rescue of cognitive function in novel object recognition (NOR) and amygdala-associated cued and contextual memory (FC).

## 2. Results

### 2.1. Increased PLD1 Levels and Increased Co-Localization with Amyloidogenic Proteins in Post-Mortem Human AD Hippocampal Slices Compared to Age-Matched Control

In our previous study, we reported that crude synaptosomal PLD1 expression (using Western blots) was increased in post-mortem AD brains compared to age-matched controls [15]. In the present study, we extend the observation further by quantifying both the increased expression of PLD1 as well as the specific co-staining between Aβ or tau with PLD1 in post-mortem AD brains (Figure 1).

We observed that the overall expression of PLD1 is significantly increased in the AD group compared to the control in both the experimental slides used for Aβ (Figure 1B, ** *p* = 0.0022) and tau (Figure 1F, ** *p* = 0.0022) co-staining with PLD1. Moreover, we report a significantly increased co-localization between PLD1-Aβ (Figure 1C, ** *p* = 0.0022) and PLD-tau (Figure 1G, * *p* = 0.0260) in AD compared to age-matched control hippocampi. Thus, the increased levels of PLD1 show an increased association with the amyloidogenic proteins implicated in the progression of AD. This is in agreement with previous studies that show astroglial PLD1 physically interacting and co-localizing with APP and caveolin-3 via the pleckstrin homology domain of PLD1 [39] and cellular studies in the neuronal N2a developmental role of PLD1 assists with the trafficking of an amyloid-precursor form of Aβ [27,61].

### 2.2. Chronic One-Month Treatment with PLD1 Inhibitor (VU0155069 or VU01 at 1 mg/kg/2 Days i.p.) in 12-Month-Old 3xTg-AD Mice Attenuates PLD1, PLD1-Aβ and PLD1-tau Co-Staining Expression Differentially in Hippocampal Subregions

We assessed whether repeated VU01 treatment was effective in reducing hippocampal PLD1 expression with attention to different subregions such as DG, CA1 and CA3 (Figure 2—*PLD1*-Aβ co-staining and Figure 3—PLD1-tau co-staining). PLD1 levels were significantly decreased in the CA1 (Figure 2J, * *p* = 0.0205; Figure 3J, * *p* = 0.0426) and DG (Figure 2B, * *p* = 0.0140; Figure 3B, ** *p* = 0.0027). The reduction in PLD1 levels did not reach significance in the CA3 (Figure 2F, *p* = 0.3357, ns; Figure 3F, *p* = 0.1079, ns). Thus, there is an overall reduction in PLD1 expression except in the CA3 region that needs further exploration. Interestingly, treatment with VU01 shows exclusive Aβ level reduction in DG (Figure 2C, *** *p* = 0.0003), but not CA1 (Figure 2K, *p* = 0.9551, ns) or CA3 (Figure 2G, *p* = 0.3357, ns). Tau expression levels remain unchanged between VU01 and saline-treated cohorts in all three regions (CA1, Figure 3K, *p* = 0.8518, ns; CA3, Figure 3G, *p* = 0.9497, ns; DG, Figure 3C, *p* = 0.7546, ns). Then we calculated the integrated density of *PLD1*-Aβ (Figure 2D,H,L) and *PLD1*-tau (Figure 3D,H,L) co-localization. We observed a significant decrease in the DG co-localization of *PLD1*-Aβ (Figure 2D, * *p* = 0.0289) and *PLD1*-tau (Figure 3D, ** *p* = 0.0087), but the other two regions CA3 and CA1 did not show significant differences between the treated and untreated cohorts.

The reduction in Aβ in DG, one of the two regions associated with neurogenesis [62,63], due to VU01 could be attributed to the activation of neuroinflammation-specific cytokines that are beneficial, since VU01 has been reported to affect inflammation in somatic tissues [64]. However, further studies will be needed to understand the specific mechanisms affected.

Collectively, these results provide evidence of the effectiveness of PLD1 inhibition with attention to regional changes using our treatment regimen. Next, we explored whether such changes were sufficient to preserve the dendritic spine integrity.

### 2.3. VU01 Dependent Chronic Inhibition of PLD1 Preserves Specific Dendritic Spine Morphologies in 12-Month-Old 3xTg-AD Mice

We previously reported in our study in 6-month-old 3xTg-AD mice treated with VU01 [42] that the mechanism of resilience involves the preservation of dendritic spine integrity. The role of Aβ on dendritic spines is well established, both using neuronal cultures [65,66] and by the loss of dendritic spines located around plaques in mouse models [67,68]. Taken together, we speculate that the repeated VU01 administration in 6-month-old 3xTg-AD mice may have protected against the Aβ-driven dendritic spine dystrophy.

While the role of tau in dendritic spine dystrophy is still emerging, there is enough evidence for its detrimental impacts resulting in clustered dendritic spine loss [9,69,70]. Therefore, we investigated how effective the repeated VU01 regimen was for preserving dendritic spine integrity at the later timeframe of 12 months in the 3xTg-AD mice (with increased tau-related insults—Figure 4). As described in our previous study, we utilized the Golgi–Cox impregnation technique [42], because of its ability to analyze spine morphology through the visualization of a low percentage of neurons (see schematic—Figure 4A). The VU01-treated cohort showed significant differences (increased numbers) in individual spine area averages (** *p* = 0.0048; Figure 4C), spine perimeter (** *p* = 0.0061; Figure 4D), filamentous spine length (** *p* = 0.0026; Figure 4E) and mushroom spine parameters of neck length (**** *p* < 0.0001; Figure 4F), neck width (**** *p* <0.0001; Figure 4G), head length (** *p* = 0.0074; Figure 4H) and head width (* *p* = 0.0101; Figure 4I) in the CA1 region of the hippocampus compared to the saline-treated 12-month-old 3xTg-AD female mice.

No significant differences were observed in spine density per 10 μm (*p* = 0.3593; Appendix A), specific spine type per 10 μm (Stubby (*p* = 0.2762; Appendix A); Filamentous (*p* = 0.0663; Appendix A); Mushroom (*p* = 0.0663; Appendix A)), number of dendritic spines (*p* = 0.3784; Appendix A) or total dendritic area measured in μm^2^ (*p* = 0.7607; Appendix A).

Thus, repeated treatment attenuating PLD1 is effective against Aβ and tau-driven clustered spine loss via the preservation of mushroom spine morphology and increased filamentous spine lengths in 12-month-old 3xTg-AD mice.

### 2.4. Restoration of Hippocampal Synaptic Function by Chronic PLD1 Inhibition Regimen in 12-Month-Old 3xTg-AD Mice

Electrophysiological assessments of high frequency stimulation-dependent long-term potentiation (HFS-LTP) and low frequency stimulation-dependent long-term depression (LFS-LTD) of hippocampal slices from the cohorts were conducted as described in the methods. In our previous studies, we demonstrated the robust recovery of the HFS-LTP by PLD1 inhibition in the hippocampal Schaffer collateral modeled using AD pathology-dependent amyloidogenic proteins (AβO or TauO) acutely in wildtype mice [15] and chronically in 6-month-old 3xTg-AD mouse models [42].

Traditionally, Aβ-associated synaptic dysfunction has been routinely assessed by measuring LTP dysfunction as a readout of functional neurodegeneration [7,12,13,14,15,42,71,72,73,74,75,76,77,78] with only relatively recent work exploring the LTD effects [75]. However, studies have clearly documented an important role for tau in LTD [79,80]. Since the 12-month-old 3xTg-AD mice are reported to show both Aβ and tau pathologies [45,46,48,50], we rationalized that the assessment of both LTP and LTD at this age will be needed to completely understand the therapeutic potential of PLD1 chronic inhibition. Furthermore, very strong correlation has been reported between clustered dendritic spine loss in AD leading to homeostatic imbalance in synaptic function that occurs via the dysregulation of synaptic depression and a corresponding compensatory excitation in brain circuits [81,82,83,84]. This could result in altered synaptic responses of long-term potentiation and long-term depression, justifying our approach to assess both LTP (Figure 5) and LTD (Figure 6) in our VU01-treated 3xTg-AD cohorts at 12 months of age.

HFS-LTP decrement observed in the 12-month-old saline-treated 3xTg-AD siblings was prevented in the VU01-treated cohort (**** *p* < 0.0001; Figure 5A,B) despite the progressive load of Aβ and tau-driven synaptic toxicity.

The improvement in synaptic function was also observed when the data were separated into males (* *p* < 0.0166; n = 5–6 mice, Figure 5C,D) and females (** *p* < 0.0015; n = 4–6 mice, Figure 5E,F) without any sex-specific difference in the inhibition effect (*p* > 0.9999 for saline (male vs. female) and *p* = 0.3491 for inhibitor (male vs. female), Figure 5G).

We also measured the input–output relationship before and after HFS-LTP (Appendix A). The input—a measure of the presynaptic fiber volley (FV)—did not show any differences when assessed for before and after stimulation (Appendix A) or between male (Appendix A) and female (Appendix A) responses (also see Appendix A). The output—the slope of the field excitatory post-synaptic potential (fEPSP)—which is also a measure of the viability of the tissue, showed no difference in the stimulation profile between the treatment groups (Appendix A) or between males (Appendix A) or females (Appendix A). It is important to note here that the slope of the response increased in both treatment groups following HFS, but the extent of the increase was pronounced in the inhibitor-treated compared to the saline-treated group (at maximum response, this was 2.3-fold for the saline-treated vs. 5.8-fold for the inhibitor-treated cohort). Qualitative trends in the post-HFS responses of female mice suggest that the efficacy of PLD1 inhibition could be better in females (Appendix A), warranting further electrophysiological assessments in future studies. The fEPSP slope as a function of FV did not reach statistical significance between the treatment groups (Appendix A) or when separated into males (Appendix A) and females (Appendix A), perhaps due to the increased variability within the saline-treated group.

We measured the paired-pulse facilitation at four different intervals (200, 100, 50 and 25 ms). We did not observe statistically significant differences between the treatment groups (Appendix A) or when the data were separated into males (Appendix A) and females (Appendix A) (also see Appendix A).

In the present study, we also looked at Schaffer collateral LTD (Figure 6). We report a significant improvement in the response of the VU01-treated 12-month-old 3xTg-AD cohort compared to the saline-treated age-matched control (**** *p* < 0.0001; n = 8–9 mice; Figure 6A,B).

When the data were separated into males (* *p* = 0.0318; n = 4–6 mice, Figure 6C,D) and females (* *p* = 0.0138; n = 3–4 mice, Figure 6E,F), we did not observe any sex-specific differences (*p* = 0.2626 for saline (male vs. female) and *p* = 0.6744 for inhibitor (male vs. female); Figure 6G).

The input–output (Appendix A) and paired-pulse measurements (Appendix A) for LFS-LTD were similar in profile to what we observed with HFS-LTP. Thus, an overall beneficial effect reflecting the homeostatic restoration of glutamatergic neurotransmission functionality is observed following our repeated VU01-associated PLD1 inhibition in 3xTg-AD mice at 12 months.

### 2.5. Chronic One-Month Treatment with PLD1 Inhibitor Ameliorates Memory Deficits in 12-Month-Old 3xTg-AD Mice

The 3xTg-AD mice injected with either inhibitor (VU01, 1mg/kg) or saline i.p. were subjected to NOR (Figure 7, see schematic in Figure 7A,B). We observed that repeated treatment with PLD1 inhibitor increased the ability of the animals to spend greater time with the novel object compared to their saline-treated sibling (Figure 7C), both at 2 h (*** *p* = 0.0003, n = 12–13 mice) and at 24 h (* *p* = 0.0211, n = 12–13 mice) after training. When the results were separated based on sex, PLD1 inhibition was effective even when the data was separated into males at 2 h (** *p* = 0.0014) and at 24 h (* *p* = 0.0404, n = 6–7 mice, Figure 7 D) or females at 2 h (* *p* = 0.0275) and at 24 h (* *p* = 0.0412, n = 6 mice, Figure 7E), suggesting equal effect of VU01 on both sexes at this age.

We did not observe any changes in the distance travelled (Appendix A) or the time that the animals were moving (Appendix A), suggesting that non-specific locomotor effects were not associated with VU01 administration. Further analysis of male-specific (Appendix A) and female-specific (Appendix A) responses also did not uncover any significant sex-specific differences.

In order to corroborate these behavioral effects and extend the observations, we performed FC behavior (Figure 8, see schematic in Figure 8A,B), where we assessed hippocampal (contextual)- and amygdala (cued)-dependent aversive associative memory [85]. The hippocampal contextual memory showed improvement in the PLD1 inhibitor-injected mice compared to the saline-treated cohort (**** *p* < 0.0001; n = 12–13 mice; Figure 8C). The improvement in response remained significant even when the cohorts were separated into males (** *p* = 0.0012; n = 6–7 mice, Figure 8F) and females (* *p* = 0.0152; n = 6 mice, Figure 8H). 

The amygdala-dependent cued response showed a significant difference between the two treatment groups (*** *p* = 0.0005; n = 12–13 mice, Figure 8D). Importantly, the pre-cue freezing was not different between the two treatments (*p* = 0.1638; Figure 8D) and was reflected even after the separation of the data into males (*p* = 0.0906; n = 6–7 mice, Figure 8G) and females (*p* = 0.9990; n = 6 mice, Figure 8I).

The cued responses showed differences between the sexes (Figure 8G,I). Female mice treated with PLD1 inhibitor showed significant cued freezing compared to their age-matched cohort injected with saline (** *p* = 0.0036; n = 6 mice, Figure 8I), but the difference did not reach significance in the male group (*p* = 0.0571; Figure 8G). However, it is important to note that the difference is not because the inhibitor-treated male mice had a lower response, rather it was because the saline-treated male group showed a higher variability (see distribution of individual animals in Figure 8G and epoch analysis in Appendix A).

When we analyzed the training sessions, we did not observe any sex-specific changes (Appendix A). Analysis of the epochs for training (Appendix A), cued (Appendix A) and contextual (Appendix A) responses corroborated the averaged responses (Appendix A). The cohorts did not show any differences in learning the behavior (Figure 8E and Appendix A).

## 3. Discussion

In the present study, we first established the scientific premise of PLD1 as a key element associated with progressive cognitive decline using human clinical samples. We demonstrated that increased PLD1 levels show increased association with Aβ and with tau—two neuropathological elements associated with AD (Figure 1). Then, we used 12-month-old 3xTg-AD mice and performed functional studies to validate whether decreasing the aberrant PLD1 was beneficial to synaptic function and underlying AD-like memory deficits. Most investigations of AD therapies have been routinely conducted only using female 3xTg-AD mice citing the prevalence of AD in females [86,87,88,89,90,91]. We have attempted to study the responses in both sexes to better interpret the underlying mechanism and the therapeutic applicability targeting the PLD1 signalosome toward the amelioration of synaptic deficits in aberrant neurological states.

There is emerging literature that the physiological levels of circulating Aβ and tau maintain normal synaptic neurotransmission in healthy states [92]. Such an observation suggests that therapeutics affecting Aβ and tau levels in diseased states should regulate, but not eliminate, Aβ and tau signaling that is important in maintaining normal cognitive functionality. Thus, we assessed how chronic VU01 treatment affects PLD1, Aβ and tau levels using immunofluorescence studies (Figure 2 and Figure 3). We report specific hippocampal regional differences in the reduction in PLD1 levels as well as PLD1-Aβ and PLD1-tau co-localization that warrants further exploration. The specificities of reduction could suggest a possible feedback loop between PLD1, Aβ and tau that results in the preservation of dendritic spines, restoration of synaptic function and prevention of cognitive decline. Additionally, there is literature that supports a role for direct interaction between PLD1 and APP [27,39,40,41,61,93] and, therefore, provides evidence for the reduction in Aβ along with PLD1 reduction by VU01 treatment is perhaps more the case rather than just being a case of “sticky” amyloidgenic protein expression loss. More elaborate studies using proximity ligation assay will be required to quantify further the importance and relevance of this association from a therapeutic point of view. Since PLD1 levels were elevated in the astrocytic and mitochondrial compartments in the previously reported study, our future studies will explore mechanisms in which PLD1 has been implicated and plays an important role in the progression of neurodegenerative states including autophagy [29,94,95,96], neuroinflammation [35,64,97,98,99,100,101,102], reactive oxygen species [103,104,105,106] and infection [35,107,108,109] and how these contribute to the synaptic dysfunction leading to cognitive decline in our studies.

It is documented that Aβ and tau progressively accumulate to toxic levels at excitatory synapses [110,111] and result in progressive dendritic spine dystrophy [112,113]. In the present study, we report that chronic PLD1 inhibition demonstrates the ability to preserve key aspects of dendritic spine morphology such as mature mushroom spine integrity and filamentous spine numbers (Figure 4), which are properties that translated to improved synaptic function (HFS-LTP and LFS-LTD) as well as underlying behaviors (NOR and FC). These findings suggest that PLD1 signaling affecting dendritic spine integrity may prove to be a key mechanism that can improve cognitive resilience at different stages in the progression of ADRD.

We assessed LTP [7,114] and LTD [115] as functional readouts of synaptic strength and plasticity (HFS-LTP—Figure 5; LFS-LTD—Figure 6) to corroborate whether the preservation of dendritic spine integrity was reflected at the synaptic level by improved functional outcomes in both these measures. We observed robust improvement without any sex-specific differences at the Schaffer collateral synapses.

Though LTP involves molecular components that affect both pre- and post-synaptic compartments, the AD literature has more reports on the Aβ and tau-related post-synaptic insults [12,92,116,117,118,119,120,121,122,123]. Many of the studies have reported the molecular changes involved in AD pathology downstream to membrane receptors in the post-synaptic compartment such as protein kinase C (PKC) [124], protein kinase M zeta (PKMζ) [125], calcium/calmodulin-dependent protein kinase II (CaMKII) [126], calcineurin (a phosphatase) [127,128] and cyclic adenosine monophosphate response element binding protein (CREB) [129], all of which affect the maintenance of dendritic spine integrity. On the other hand, glutamate neurotransmission associated LTD that is affected in AD by either Aβ [130] or tau [80], has opposite effects on the above-mentioned signaling cascade and could, therefore, be instrumental in dendritic spine retraction. Thus, it is proposed that LTD in healthy individuals works (along with LTP) to regulate the dendritic spine morphology and enable “forgetting” [131]. Therefore, in theory, Aβ and tau could hijack and compromise LTP, exacerbate LTD and in turn cause “forgetting” via the reduction in dendritic spines [132]. Perhaps repeated PLD1 inhibition at 1 mg/kg may be optimal in modulating glutamatergic neurotransmission to preserve synaptic functionality, thereby altering the homeostasis enough to resist cognitive decline without affecting the physiological functionality associated with Aβ and tau. Taken together, our observations support a possibility that VU01 seems to have a much larger impact on synaptic function and plasticity than on spine morphometric properties (i.e., for most synapse morphometric properties, the magnitude of difference between saline and inhibitor is fairly small, despite significant statistical comparisons). Thus, VU01-mediated inhibition may also affect the biochemical mechanisms of synapse function/plasticity (e.g., glutamate receptor function, balance of protein kinase/phosphatase activities independently of spine morphology). Alternatively, it is possible that there are extrasynaptic effects of VU01 that are responsible for the multimodal improvement that we report here as we have elaborated earlier in the discussion section. Of greater importance to this particular study, where we have attempted to address therapeutic efficacy, we chose to compare the effects of VU01 in the pathological background rather than age-matched wildtype control because we were not addressing the mechanistic changes at this level of study, which would require more extensive and exhaustive analysis that is beyond the scope of this study, but would be addressed in a future study.

The 3xTg-AD triple transgenic model of AD-like neuropathology considers (1) age-dependent increase and (2) sex-dependent effects with progressively accumulating Aβ and tau, thus mirroring the human pathological profile [43,44,52]. Familial early-onset Alzheimer’s disease (FEOAD) or early-onset AD (EOAD) associated with dominant mutations in *APP*, *PSEN1* and *PSEN2* have a high penetrance for displaying many of the clinical symptoms associated with AD. Thus, characterized in 2003, the 3xTg-AD mouse model is a popular pre-clinical AD model [52], with notable AD-related pathologies which are faithfully replicated. Aβ plaques are first detected at 6 months in the hippocampus that accumulate progressively with age, such as happens in humans. Moreover, cortical Aβ plaques appear at 12 months and also exhibit an age-dependent progression and this was reported initially [52] and replicated recently [47]. Most important, for our current study, tau phosphorylation was observed from 12 months in the hippocampus, particularly in the pyramidal neurons of the CA1 region [47,52]. Thus, while it is important to note here that this extensively used mouse model of AD is unique among many to recapitulate Aβ and tau-driven neurotoxicity found in human tissue or imaging experiments, and vice versa, successful therapies developed and tested have been universally unsuccessful in human clinical trials, prompting a reassessment of the development, use and interpretations of the data acquired from such models [133]. One critical factor that is often disregarded in therapeutic assessment studies at the pre-clinical level is the age. Our present systematic study at 12 months addresses exactly that component of therapeutic importance that is missing in other studies using this remarkable transgenic model that shows the progressive advancement of synaptic dysfunction and associated cognitive decline. Thus, our study is novel in that it emphasizes the observation that a repeated dose of the therapeutic even later in the progression remains efficacious, an observation that is more unique than common and needs to be a gold standard to study the effects of the drug, using multimodal approaches at more than one age to illustrate the robustness of the intervention before proceeding to human clinical trials.

Another important novelty that emerges from our study is the potential of PLD1 therapeutics in late-onset AD (LOAD). The search for an “ideal” transgenic model of late-onset AD (LOAD) is a key requirement for the NIH/NIA-established Model Organism Development and Evaluation of Late-onset Alzheimer’s Disease (MODEL-AD) [134]. Under this premise, the LaFerla group conducted a systematic assessment and revisited the adequacy for the 3xTg-AD model recently [133]. They concluded that the slow pathology development in an age-specific and sex-specific manner will be advantageous to study the interaction of amyloidogenic insults on synaptic dysfunction associated with LOAD. Importantly, LOAD is linked to a number of variants with the most well-known occurring in the ε4 allele of apolipoprotein E (APOE4) that result in the disruption of lipid homeostasis [135,136].

APOE occurs in three isoforms—E2, E3 and E4—which differ in a single amino acid change. E4 allele increases the risk of AD in a dose-dependent manner compared to E3 allele, while E2 is reportedly protective with its biological role remaining understudied. It facilitates the binding and uptake of lipoprotein complexes via low-density lipoprotein receptors (LDLRs), or lipoprotein receptor 1 (LRP1). Multiple studies point to the importance of APOE as a universal biological variable along with age and gender in the progression of AD. APOE4 is the major genetic risk factor for developing sporadic AD (sAD), where 1 copy confers a 3-fold increase while 2 copies increase the risk to 12-fold. Female APOE4 carriers have an increased AD risk compared to male APOE4 carriers. Recent transcriptomic studies of human APOE3 and APOE4 knockin mice (APOE3-KI and APOE4-KI resp.) from 5 to 20 months of age show prominent changes in neuronal calcium signaling and synaptic function, implicating the phospholipase D (PLD) pathway [137]. APOE is the most abundant brain lipoprotein chaperones expressed mainly in astrocytes. Combined with the observation that aberrant upregulated PLD1 expression and activity was reported in reactive astroglial cells from AD brains, where a physical interaction with amyloid precursor protein (APP) [39] alters the mitochondrial function in the brains of scrapie-infected mice [40] as well as in AD brains [41], we speculate that attenuating PLD1 using VU01 in 12-month-old 3xTg-AD may impinge on APOE-related dyslipidemia that may be another key mechanism that needs to be explored to realize the full therapeutic potential of VU01 as a potential LOAD intervention.

Heavier Aβ burden in female 3xTg-AD [138,139,140,141] correlated with a greater impairment in spatial reorientation [139], compared to male mice. Higher levels of tau are observed in older (18–20 month old) females compared to males [48]. It is important to note here that we have separated the male and female responses for understanding the sex-specific effects of PLD1 inhibition. However, we will further explore these aspects in the future with more numbers for a better powered study. Nevertheless, our stratified approach and a multimodal assessment provide a degree of confidence in conducting such sex-specific assessments that is important for properly addressing the therapeutics and mechanisms in neurodegenerative states that are known to have gender-specific effects. 

In our previous study [42], no sex-specific differences were observed within the saline- or inhibitor-treated or between the two treatment groups in the amygdala-dependent aversive cued-fear memory response despite detectable levels of intraneuronal and extracellular Aβ in the basolateral/lateral amygdala reported at this age [44]. Robust tau expression in the amygdala of 3xTg-AD mice has been reported at later ages beginning from 9 months through 26 months [51]. Therefore, we speculated that the combined effect of Aβ and tau in 12-month-olds would be more detrimental than the effect of Aβ alone at 6 months in impairing the robust amygdala-dependent cued (fear-conditioned) memory. Indeed, our results (Figure 8) indicate that the 12-month-old 3xTg-AD saline-treated animals have a poor cued memory that is rescued by treatment with the inhibitor. Our sex-stratified approach also uncovered that the inhibitor-treated male cohort failed to reach significance when compared to the saline-treated male group for the cued response. Surprisingly, no intra-treatment differences were noted between the male and female inhibitor-treated cohort for the cued response (Figure 8). This lack of statistical significance may be due to the variability in the saline-treated male cohort (see Figure 8G). An earlier study had reported increased anxiety in 12-month and 15-month-old 3xTg-AD male mice [142] compared to females. Other differences between 2 months to 15 months are attributed to neuroendocrine differences [48] that result in males showing more errors and more latency in memory acquisition [46,143]. Taken together, these results indicate that aging-related, sex-specific endocrine signaling contribute to the variation in male behavioral response seen in our saline-treated male cohort. It is, therefore, remarkable, and pertinent that regardless of such variations, the efficacy of PLD1 inhibition was not different in the male and female mice within the repeated VU01-treated group. Therefore, we speculate that the PLD1 inhibition (by VU01 at 1mg/kg every alternate day for a month regimen) preserves amygdala memory mechanisms by possibly regulating the hormonal effects on synaptic function in males.

In the NOR study (Figure 7), no sex-specific difference was observed within the saline-treated or the inhibitor-treated cohorts. It has been reported in human clinical studies that the volume of the total perirhinal cortex in AD patients is decreased compared to control subjects, but without any gender differences [144]. Short-term storage of object memory occurs in the perirhinal cortex before being transferred to the hippocampus for consolidation [145]. Thus, we speculate that the lack of sex differences in our NOR study is because NOR heavily relies on perirhinal cortex function.

Thus, in this study, we find that chronic PLD1 inhibition (1 mg/kg VU01 once every 2 days i.p. for 30 days) at later stages of AD-like neuropathology, that has a greater dependence on tau, (with additional effects on neuroinflammation as well as astroglial mechanisms impinging on APOE-related dyslipidemia that need to be explored further) was sufficient to reduce PLD1 expression and association with amyloidogenic proteins, preserve dendritic spine morphology, improve hippocampal synaptic function and rescue hippocampal and amygdala-associative memory deficits.

## 4. Materials and Methods

### 4.1. Drugs

PLD1 inhibitor (VU0155069) was obtained from Tocris Bioscience (Bio-Techne, Minneapolis, NE, USA).

### 4.2. Human Samples

Post-mortem brain tissues were a gracious gift from Dr. Giulio Taglialatela, PhD, available to us as a member of the Mitchell Center for Neurodegenerative Diseases through his capacity as the Director for the Center. The source of the tissues is the Oregon Brain Bank at Oregon Health and Science University (OHSU; Portland, OR, USA). Briefly, donor subjects of either sex were enrolled and clinically evaluated in studies at the National Institutes of Health-sponsored C. Rex and Ruth H. Layton Aging and Alzheimer’s Disease Center (ADC) at OHSU, via the OHSU Institutional Review Board (IRB). Informed consent was obtained from all participants before their enrollment in the studies at the ADC in brain-aging and for receiving annual neurologic and neuropsychological evaluations. Following neuropathological assessments post-mortem, a neuropathologist-scored database was generated for each brain tissue, according to standardized CERAD (Consortium to Establish a Registry for Alzheimer’s Disease) criteria and Braak staging.

The cases used in this study are described in Table 1. To ensure that the variations in post-mortem interval (PMI) did not affect any measurements, we have provided a correlation analysis between PMI values and the immunofluorescence (IF) performed for PLD1. No correlation was found (Appendix A), and, therefore, observed differences could not be attributed to differences in non-specific post-mortem tissue degradation. It is nonetheless important to appreciate that five of the brains obtained following 10 h PMI might not necessarily fully reflect freshly obtained brain tissue.

### 4.3. Animals

This study was conducted in accordance with the National Research Council’s “Guide for the Care and Use of Laboratory Animals (8th Edition)” in the animal care facility at The University of Texas Medical Branch at Galveston (UTMB) which is accredited by the Association for Assessment and Accreditation of Laboratory Animal Care (AALAS), International. All procedures were approved by the Institutional Animal Care and Use Committee (IACUC) and were performed according to the National Institutes of Health (NIH) Guidelines [146] on the use of laboratory animals.

Male and female 3xTg-AD transgenic mice were purchased from Jackson Labs (Bar Harbor, ME, USA) and maintained through a breeding program at UTMB. Mice were housed—five per cage in their filter-top cages in a temperature-controlled environment at 22 °C, humidity 40% and a 12:12 h light–dark cycle, with regular chow provided ad libitum. We had to utilize three cohorts of the 3xTg-AD mice for the experiments described in this study [Round 1: 6 females and 7 males (VU01 injected) and 6 females and 6 males (0.9% saline); Round 2: 5 females (VU01 injected) and 5 females (0.9% saline)], separated by few months to complete the experiments described here. Round 3: 4 males and 4 females (2 of each sex injected with VU01 or saline) separated by 7 months from second round. Thus, we had n = 20 animals total for VU01-injected and n = 21 animals total for saline- or vehicle-injected to conduct the behavioral, electrophysiological and dendritic spine studies. Animals (aged ~11 months) received a single injection intraperitoneally (i.p.) of 1 mg/kg of VU01 diluted in 0.9% saline solution (inhibitor-treated cohort) or an equivalent amount of 0.9% saline (saline-treated cohort) and returned to their cage. This was repeated every alternate day (alternating the injection side to reduce incidence of repetitive injection-related inflammation), for a period of one month. Based on previous reports, the results were also analyzed separately for males and females. All behavioral testing was performed within the 12 h light cycle for return to home cages prior to the 12 h dark cycle. After completion of the behaviors, the animals were processed as described under field electrophysiological recordings section. The second cohort of animals injected for Golgi staining were also deeply anesthetized with isoflurane, and immediately the brain was extracted from the skull, washed with phosphate buffered saline pH 7.4 (ThermoFisher Scientific, Waltham, MA, USA) and processed as described under Tissue Processing and Golgi Staining section. 

### 4.4. Field Electrophysiological Recordings

Our standard protocol was used as previously described [13,14,15,42,76,77,147,148]. Briefly, mice were deeply anesthetized with isoflurane and transcardially perfused with ~30 mL of room temperature carbogenated (95% O_2_ and 5% CO_2_ gas mixture) NMDG-artificial cerebrospinal fluid (aCSF) (in mM-93 N-Methyl-D-Gluconate, 2.5 KCl, 1.2 NaH_2_PO_4_, 30 NaHCO_3_, 20 C_8_H_18_N_2_O_4_S, 25 C_6_H_12_O_6_, 5 C_6_H_7_O_6_Na, 2 CH_4_N_2_S, 3 C_3_H_3_NaO_3_, 10 MgSO_4_,7H_2_O, 0.5 CaCl_2_,2H_2_O, 12 C_5_H_9_NO_3_S, pH 7.4) and sliced using Compresstome VF-300 (Precisionary Instruments, Greenville, NC, USA) in carbogenated NMDG-aCSF to obtain 350 μm transverse brain sections. Slices were allowed to recover for 10 min in carbogenated NMDG-aCSF at 33 °C. Slices were then maintained at room temperature in a modified carbogenated HEPES holding aCSF solution (in mM-92 NaCl, 2.5 KCl, 1.2 NaH_2_PO_4_, 30 NaHCO_3_, 20 C_8_H_18_N_2_O_4_S, 25 C_6_H_12_O_6_, 5 C_6_H_7_O_6_Na, 2 CH_4_N_2_S, 3 C_3_H_3_NaO_3_, 2 MgSO_4_,7H_2_0, 2 CaCl_2_,2H_2_0, 12 C_5_H_9_NO_3_S, pH 7.4). Slices were recorded in carbogenated standard recording naCSF (in mM-124 NaCl, 2.5 KCl, 1.2 NaH_2_PO_4_, 24 NaHCO_3_, 5 C_8_H_18_N_2_O_4_S, 13 C_6_H_12_O_6_, 2 MgSO_4_,7H_2_0, 2 CaCl_2_,2H_2_0, pH 7.4). Evoked field excitatory post-synaptic potential (fEPSP) recordings were performed by stimulating the Schaffer collateral pathway (located in stratum radiatum) using a stimulating electrode of ~22 kΩ resistance placed in the CA3 region and glass recording electrodes in the CA1 region. Current stimulation was delivered through a digital stimulus isolation amplifier (A.M.P.I., ISRAEL) and set to elicit a fEPSP approximately 30% of maximum for synaptic potentiation experiments using platinum–iridium-tipped concentric bipolar stimulating electrodes (FHC Inc., Bowdoin, ME, USA). The use of platinum–iridium wire and diphasic pulses can help minimize electrode polarization [149]. Using a horizontal P-97 Flaming/Brown Micropipette puller (Sutter Instruments, Novato, CA, USA), borosilicate glass capillaries were used to pull recording electrodes and filled with naCSF to obtain a resistance of 1–2 MΩ. Field potentials were recorded in CA1 stratum radiatum using a Ag/AgCl bridge with CV7B headstage (Molecular Devices, Sunnyvale, CA, USA) located ~1–2 mm from the stimulating electrode. LTP was induced using a high frequency stimulation protocol (3 × 100 Hz, 20 s) as previously described [13,14,15,42,76,77,147]. LTD was induced using a low frequency stimulation protocol (900 × 1 Hz). Recordings were digitized with Digidata 1550B (Molecular Devices, Sunnyvale, CA, USA), amplified 100× and digitized at 6 kHz using an Axon MultiClamp 700B differential amplifier (Molecular Devices) and analyzed using Clampex 10.7 software (Molecular Devices) as previously described [148]. To assess basal synaptic strength, 250 μs stimulus pulses were given at 10 intensity levels (range, 100–1000 μA) at a rate of 0.1 Hz. Three field potentials at each level were averaged, and measurements of fiber volley (FV) amplitude (in millivolts) and fEPSP slope (millivolts per millisecond) were performed using Clampfit 10.7 software. Synaptic strength curves were constructed by plotting fEPSP slope values against FV amplitudes for each stimulus level. Baseline recordings were obtained for 10 min by delivering single pulse stimulations at 20 s intervals. All data are represented as a percentage change from the initial average baseline fEPSP slope obtained for the 10 min prior to HFS. Maximum of two slices were recorded per animal and averaged to give the response per animal.

### 4.5. Behaviors

#### 4.5.1. Novel Object Recognition

NOR was performed as described previously [13,14,15,42,76]. Briefly, animals were habituated for two consecutive days and assessed for normal locomotion and acclimation to the test environment (see schematic in Figure 7). After placement in the open field box for two 10 min test sessions that were 24 h apart, the AnyMaze (Stoelting Co., Wood Dale, IL, USA) video tracking software was used to quantify various locomotor parameters: total distance traveled, time spent moving >50mm/s, number of rears, number of entries into and time spent in the center 1/9th of the locomotor arena. Twenty-four hours after the last habituation session, animals were subjected to training in a 10 min session of exposure to two identical, non-toxic objects in the open field box. The time spent exploring each object was recorded using an area 2 cm^2^ surrounding the object and was defined such that nose entries within 2 cm of the object were recorded as time exploring the object. After the training session, the animal was returned to its home cage. After a retention interval of 2 h and subsequently 24 h, the animal was returned to the arena in which two objects, one identical to the familiar object but previously unused (to prevent olfactory cues and prevent the necessity to clean objects during experimentation) and one novel object. The animal was allowed to explore for 10 min, during which the amount of time exploring each object was recorded. Objects were randomized and counterbalanced across animals. The animals were returned to their home cages with food/water ad libitum for 24 h minimum. After the rest period of minimum three days, the animals were tested for fear conditioning as described below. For novel object recognition tests, the percentage time exploring each object (familiar versus novel) is reported as an object discrimination index (ODI). An index above 0.5 is indicative of novelty associated with the object. Each mouse was tested at 2 h and at 24 h with the intention of assessing the shorter and longer time frames in memory recall. Different novel objects (color and shape) were used in the 24 h test compared to the 2 h test, to avoid performance deficits.

#### 4.5.2. Fear Conditioned Response

Contextual and cued fear-conditioned responses were assessed using our standard two-pairings fear-conditioning training protocol as previously described [42,150], utilizing the UTMB Rodent In Vivo Assessment Core Facility. Briefly, the standard protocol consisted of a training phase when the mice were placed in a particular environment—a standard mouse fear-conditioning chamber (Med Associates, Fairfax, VT, USA, a training chamber with lighting, geometry, odor that constitutes the context conditioned stimulus, CS) and allowed to explore for 3 min. An auditory CS (80 dB white noise) was then presented for 30 s and one footshock (0.8 mA, 2 s duration; the unconditioned stimulus, US) delivered during the last 2 s of the auditory CS. A second presentation of the auditory CS and the US was delivered at the 5 min mark and the animals then left in the cage for another 2 min. Twenty-four hours later, the mice were returned to the same training chamber and the context test for fear learning performed. The amount of freezing the mice exhibited during five minutes in the training chamber was measured. Between two to four hours later, the cued test was performed in a completely novel context. The animals were placed in the testing chamber and freezing was measured for three minutes before the auditory CS was represented and freezing quantified over the next three minutes. Freezing was quantified using FreezeFrame automated video capture and software analysis (Coulbourn Instruments, Whitehall, PA, USA) and evaluated as percentage freezing in 30 s (training) or 60 s (contextual, cued) bins. Epochs were averaged to provide the data as number of animals per group.

### 4.6. Tissue Processing and Golgi–Cox Staining

As previously reported, brain hemispheres, obtained as described in the animals section, were stained using the FD Rapid Golgi Stain Kit (PK401, FD Neurotechnologies, Columbia, MD, USA) and according to the manufacturer’s instructions [42]. Tissue slices were impregnated in chromate mixture of Solution A (potassium dichromate and mercuric chloride) and Solution B (potassium chromate). After 24 h, chromate solution was replaced gently without disturbing the tissue, and then left in dark for 15 days. Subsequently, tissue slices were immersed in Solution C for 24 h. After 24 h, Solution C was replaced, according to the manufacturer’s instructions. These brains were sliced in 30 μm sections, mounted three per slide on gelatin-coated slides, sequentially for each animal by FD Neurotechnologies. For further microscopic assessments, the slides were shipped back to our lab after being processed by FD Neurotechnologies. Slides were stored in darkness.

### 4.7. Dendrite Imaging

Previously published criteria and standards for dendritic imaging were used [151,152]. A blinded experimenter performed imaging and further analysis of the slides. An All-in-One Fluorescence Microscope BZ-X800E (Keyence Corporation of America, Itasca, IL, USA) was used to image Golgi-stained neurons at high magnification (100X oil-immersion objective) using the brightfield options available in the microscope. It was possible to determine the morphology of individual spines by magnifying using the 3X optical zoom option and subsequently quantified. To cover the full depth of the dendritic arbors (20–30 μm), Z-stack images were collected at 0.3 μm intervals and then compressed into a single TIFF image using the BZ-H4A software. ImageJ software was utilized (Open Source from National Institutes of Health, Bethesda, MD, USA) on these TIFF stacks for subsequent quantitative analysis. For each animal, three hemispheric slices per slide with the best representation of the Schaffer collateral were chosen for quantification. From each tissue slice, five distinct areas were imaged and analyzed. The following criteria were used to select areas for imaging: (1) located centrally within the tissue sample depth, (2) not obscured by large staining debris and (3) fully impregnated. If the areas met the criteria, a single dendritic length was imaged per area. For dendrite selection, the following criteria were used: (1) unobstructed/isolated/not overlapping other dendrites, (2) length > 30 µm and (3) diameter approximately 1 µm. If more than two dendrites fulfilled the criteria from a single cell, the first dendrite clockwise was the only dendrite selected. Each tissue slice was initially imaged under low 20× magnification to establish the regions of interest and to determine the five distinct areas for dendrite selection.

### 4.8. Immunofluorescence

Fresh frozen tissue blocks from human (n = 6 per group) or mice (n = 4 per group, 2 males and 2 females) were removed from storage at −80 °C, equilibrated at −20 °C, embedded in O.C.T. (optimal cutting temperature) compound (Tissue-tek). Sections as thick as 12 µm were captured onto Superfrost Plus slides (ThermoFisher Scientific). Prepared slides were stored at −20 °C until use. Slides were fixed in 4% paraformaldehyde in 0.1 M PBS, pH 7.4 for 30 min at room temperature (RT). Non-specific binding sites in the sections were blocked with 5% bovine serum albumin (BSA, ThermoFisher Scientific), 10% normal goat serum (NGS, Millipore Sigma, Burlington, MA, USA). For permeabilizing the sections, 0.5% Triton X-100 and 0.05% Tween-20 for 1 h were used at RT. Slides were incubated with primary antibodies diluted in PBS containing 1.5% NGS with 0.25% Triton X-100 overnight at 4 °C. Following primary antibodies were used: mouse monoclonal Aβ (1:100, 4G8 or Purified anti-β-Amyloid, 17–24 Antibody (Previously Covance catalog #SIG-39220) #800712, Biolegend, RRID:AB_2734548), rabbit anti-PLD1 (1:200, catalog #ab50695, Abcam, RRID, AB_2237051), mouse monoclonal anti-tau 5 for human tissues (1:2000, catalog #806404, Biolegend, RRID: AB_2715857) or chicken polyclonal anti-tau (1:200, catalog #ab75714, RRID: AB_1310734). Before incubation with the Alexa Fluor-conjugated secondary antibodies, slides were washed in PBS 3 times, 10 min each. Following secondary antibodies were used: goat anti-rabbit Alexa Fluor 488 (1:400, catalog #A-11034, RRID:AB_2576217), goat anti-mouse Alexa Fluor 594 (1:400, catalog #A-11032, RRID:AB_2534091), goat anti-chicken Alexa Fluor 594 (1:400, catalog #A11039, RRID:AB_2534099) diluted in PBS containing 1.5% NGS with 0.25% Triton X-100 for 1 h at RT. Finally, slides were washed in PBS 3 times, 10 min each, then treated with 0.3% Sudan Black in 70% EtOH for 10 min to block autofluorescence caused by lipofuscin, washed again with deionized water and coverslipped using fluoromount-G-containing 4′,6′-diamidino-2-phenylindole dihydrochloride (DAPI, SouthernBiotech, Birmingham, AL, USA).

### 4.9. Quantitative Microscopy

Images were acquired either with Keyence BZ-X800 microscope, by using 10× and 60× (oil immersion) objectives or IX83 Confocal microscope (Olympus Corporation, Tokyo, Japan) using 40× and 60× (both oil immersion) objectives for all the immunoreacted sections. For each subject, 2 sections were analyzed and 5 images per section were captured (for the mouse slices, 5 images per region of CA1, CA3 and DG were captured). Resolution was kept at 1920 × 1440 pixel with a Z-step of 2 at 12 µm thickness. All layers from a single image stack were projected on a single slice (stack/Z projection) to increase the confidence of co-localization and quantification. Quantitative analysis was performed using ImageJ software (downloaded from NIH; http://imagej.nih.gov/ij), and the intensity of fluorescence for each marker was analyzed as integrated density to account for overall distribution. The co-localization between two markers was evaluated and quantified using the Pearson’s correlation coefficient and Mander’s correlation coefficient in ImageJ. Representative images were composed in an Adobe Photoshop CC2020 format.

### 4.10. Statistics

All data are reported as mean ± SEM. Statistical significance was calculated using GraphPad Prism 9.2 (San Diego, CA, USA). All statistical tests were two-tailed, with the threshold for statistical significance set at 0.05. To account for non-normal distribution of data, either non-parametric *t*-tests (Mann–Whitney U or Wilcoxon rank sum) or one-way ANOVA (Kruskal–Wallis test) followed by Geisser–Greenhouse correction for mixed effects analysis, Holm–Sidak multiple comparisons test with individual variances computed for each comparison or uncorrected Fisher’s LSD with individual variances computed was applied as appropriate to account for variability of differences. Double blinding was performed—this was achieved by one scientist performing the dilutions and providing it to the experimenter conducting the injections and subsequent experiments with a code denoting the different treatments. Once the analysis was completed, the code was broken.

## 5. Conclusions

We previously identified a key mechanism that links increased synaptosomal PLD1 to dendritic spine dystrophy resulting in compromised synaptic function. We also showed that the cognitive decline (driven predominantly by Aβ) in 6-month-old 3xTg-AD was attenuated by a repeated VU01 dose. Here, we increased the thorough and multidisciplinary approach to validate the ability of the therapeutic design and we attempted to address the consistency of outcomes at progressive stages of neurodegenerative insult. Our results suggest that there is efficacy of the same regimen in preventing combined progressive insults of Aβ and tau-driven synaptic dysfunction. We also paid attention to treatment-related sex-specific differences, an important consideration in advancing the therapeutics to human clinical stages. Taken together, we find that repeated PLD1 inhibition is a viable therapeutic approach not only at early but also later stages in the progression of ADRD-like synaptic dysfunction and underlying memory deficits since it confers resilience to dendritic spine dystrophy. Additionally, we report, for the first time using immunofluorescence approaches, that there is a distinct decrease in PLD1 expression (increasing our confidence in the brain-penetrant effects of VU01 functionally) that follows a hippocampal subregion preponderance with effects on Aβ and tau with PLD1 co-localization that warrants deeper assessment of protein–protein interaction in future studies that may provide critical insights regarding the success of our therapeutic approaches. Thus, a continued systematic approach incorporating the above aspects will be critical in addressing the complementary nature of PLD1 therapeutic intervention in complementing the immunosuppression therapies against amyloidogenic proteins in effectively preventing the progression of neurodegenerative states at any clinical stage of the disease.

## Figures and Tables

**Figure 1 ijms-24-03372-f001:**
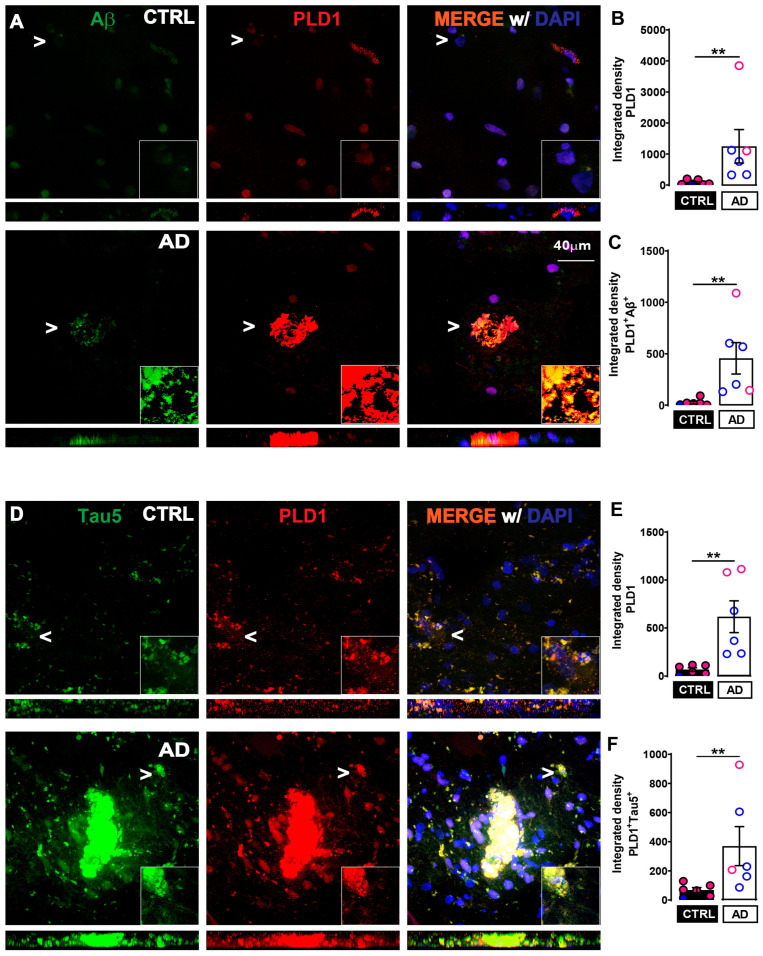
Increased co-staining of PLD1 and Aβ or tau in post-mortem AD hippocampi compared to CTRL. (**A**) Double immunofluorescence showing higher levels of PLD1 (red) with Aβ (green) in AD samples compared to age-matched controls. (**B**) Increased PLD1 expression in AD (** *p* = 0.0022). (**C**) Significantly increased co-localization of PLD1 and Aβ in AD (** *p* = 0.0022). (**D**) Double immunofluorescence showing higher levels of PLD1 (red) with tau (green) in AD samples compared to age-matched controls. (**E**) Increased PLD1 expression in AD observed with PLD1-tau5 co-staining (** *p* = 0.0022). (**F**) Significantly increased co-localization of PLD1 and tau in AD (** *p* = 0.0260). Merge (yellow) is shown with DAPI staining of nuclei in the representative panels. White arrowheads denote the region that is zoomed at the lower right-hand corner to better demonstrate the interaction. At the bottom of each square panel, a rectangular panel of the confocal stacks shows the qualitative distribution and co-localization. Values are expressed as mean ± SEM. n = 6 subjects per group. Unpaired non-parametric Mann–Whitney test. Scale bar = 40 μm. Each circle in the graph represents a single subject. Red/pink circles indicate females while blue circles indicate males.

**Figure 2 ijms-24-03372-f002:**
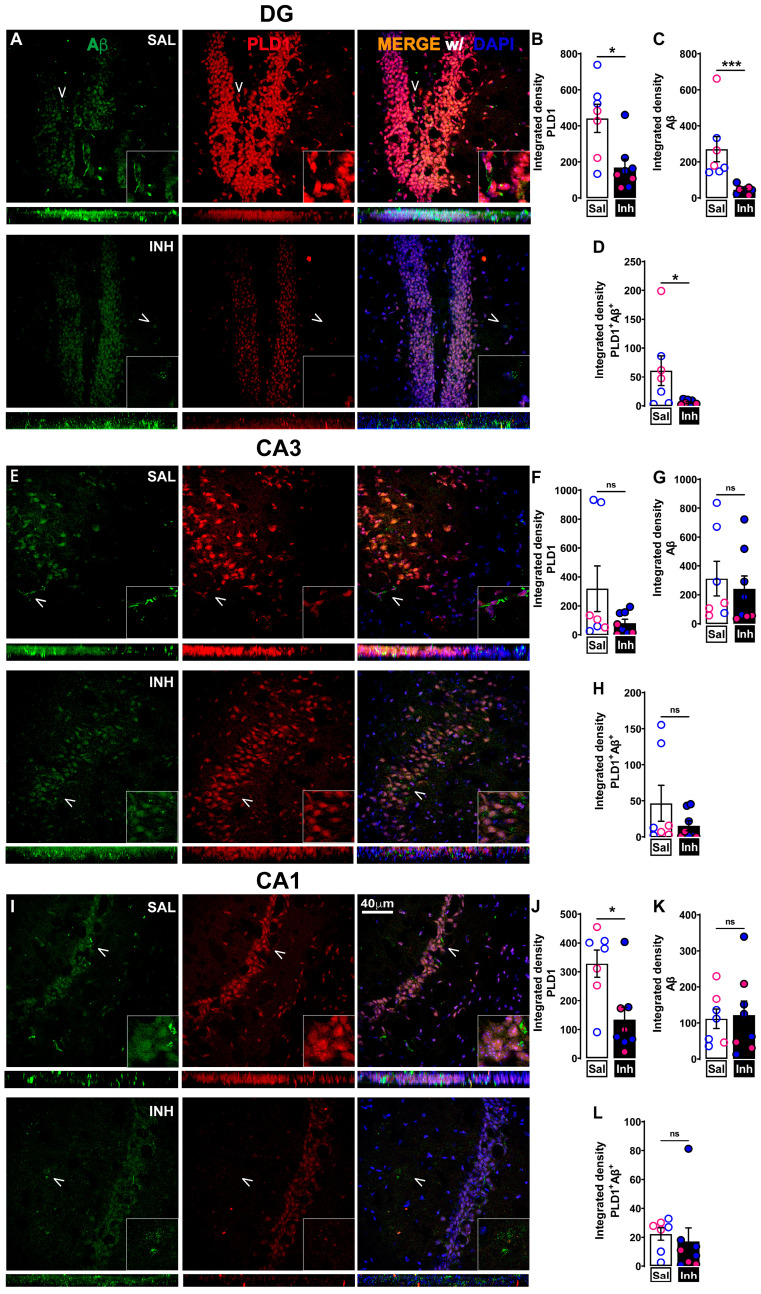
VU01 treatment (1 mg/kg/2 days for a month) in 12-month-old 3xTg-AD mice attenuates PLD1-Aβ expression differentially in hippocampal subregions. (**A**) Double immunofluorescence showing reduced co-staining of PLD1 (red) with Aβ (green) following repeated inhibitor treatment (INH) compared to age-matched saline (SAL)-treated 3xTg-AD cohort in the dentate gyrus (DG). (**B**) Decreased PLD1 expression in INH compared to SAL in DG (* *p* = 0.0205). (**C**) Decreased Aβ expression in INH compared to SAL in DG (*** *p* = 0.0003). (**D**) Significantly decreased co-localization of PLD1 and Aβ in INH compared to SAL in DG (* *p* = 0.0289). (**E**) Double immunofluorescence showing no significant reduction in co-staining of PLD1 (red) with Aβ (green) in INH compared to SAL in CA3. (**F**) PLD1 expression does not show significant decrease in INH compared to SAL in CA3. (**G**) Aβ expression does not show significant decrease in INH compared to SAL in CA3. (**H**) Co-localization of PLD1 and Aβ does not show significant decrease in INH compared to SAL in CA3. (**I**) Double immunofluorescence showing no significant reduction in co-staining of PLD1 (red) with Aβ (green) in INH compared to SAL in CA1. (**J**) Significant PLD1 decrease in INH compared to SAL in CA1 (* *p* = 0.0205). (**K**) Aβ expression does not show significant decrease in INH compared to SAL in CA1. (**L**) Co-localization of PLD1 and Aβ does not show significant decrease in INH compared to SAL in CA1. Merge (yellow) is shown with DAPI staining of nuclei in the representative panels. White arrowheads denote the region that is zoomed at the lower right-hand corner to better demonstrate the interaction. At the bottom of each square panel, a rectangular panel of the confocal stacks shows the qualitative distribution and co-localization. Values are expressed as mean ± SEM. n = 4 subjects per group (2 males and 2 females)—total of 7–8 slices from these animals were used to quantify the results. Unpaired non-parametric Mann–Whitney test. Scale bar = 40 μm. Red/pink circles indicate females while blue circles indicate males. ns, no significance.

**Figure 3 ijms-24-03372-f003:**
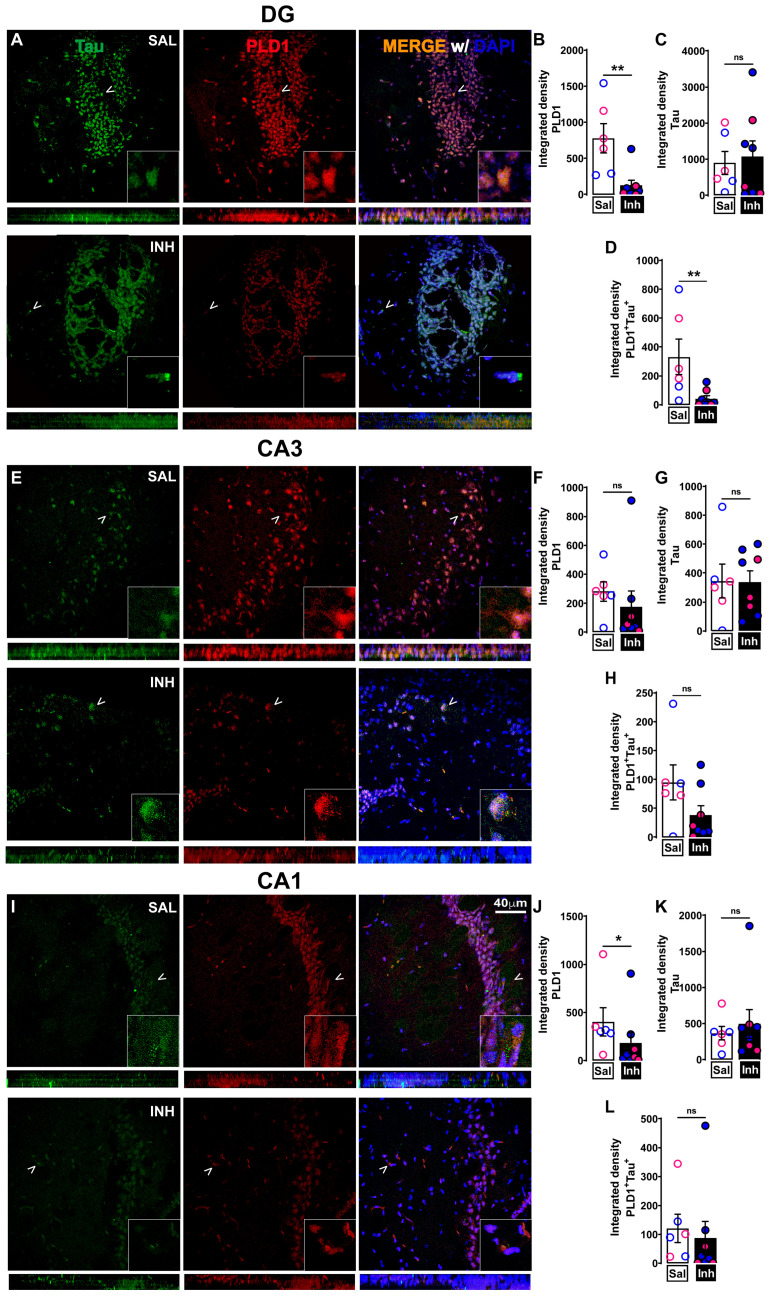
Repeated VU01 (1 mg/kg/2 days for a month) in 12-month-old 3xTg-AD mice attenuates PLD1-tau expression differentially in hippocampal subregions. (**A**) Double immunofluorescence showing reduced co-staining of PLD1 (red) with tau (green) following repeated inhibitor treatment (INH) compared to age-matched saline (SAL)-treated 3xTg-AD cohort in the dentate gyrus (DG). (**B**) Decreased PLD1 expression in INH compared to SAL in DG (** *p* = 0.0027). (**C**) Tau expression does not change between INH and SAL in DG. (**D**) Significantly decreased co-localization of PLD1 and tau in INH compared to SAL in DG (** *p* = 0.0087). (**E**) Double immunofluorescence showing no significant reduction in co-staining of PLD1 (red) with tau (green) in INH compared to SAL in CA3. (**F**) PLD1 expression does not show significant decrease in INH compared to SAL in CA3. (**G**) Tau expression does not show significant decrease in INH compared to SAL in CA3. (**H**) Co-localization of PLD1 and tau does not show significant decrease in INH compared to SAL in CA3. (**I**) Double immunofluorescence showing no significant reduction in co-staining of PLD1 (red) with tau (green) in INH compared to SAL in CA1. (**J**) Significant PLD1 decrease in INH compared to SAL in CA1 (* *p* = 0.0426). (**K**) Tau expression does not show significant decrease in INH compared to SAL in CA1. (**L**) Co-localization of PLD1 and tau does not show significant decrease in INH compared to SAL in CA1. Merge (yellow) is shown with DAPI staining of nuclei in the representative panels. White arrowheads denote the region that is zoomed at the lower right-hand corner to better demonstrate the interaction. At the bottom of each square panel, a rectangular panel of the confocal stacks shows the qualitative distribution and co-localization. Values are expressed as mean ± SEM. n = 4 subjects per group (2 males and 2 females)—total of 7–8 slices from these animals were used to quantify the results. Unpaired non-parametric Mann–Whitney test. Scale bar = 40 μm. Red/pink circles indicate females while blue circles indicate males. ns, no significance.

**Figure 4 ijms-24-03372-f004:**
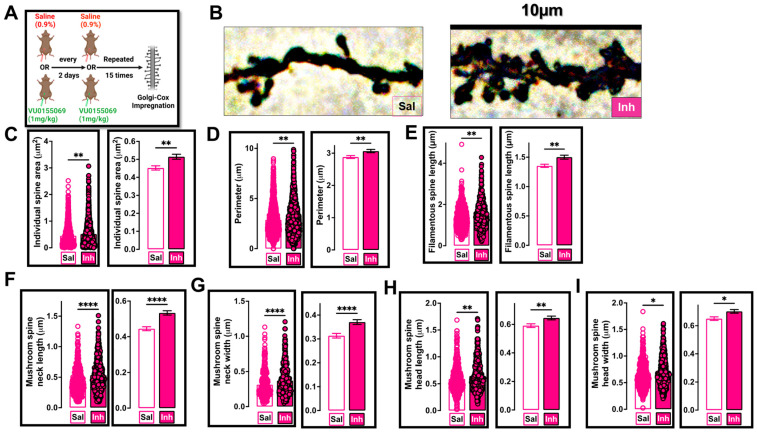
PLD1 inhibition in 12-month-old 3xTg-AD mice prevents progressive Aβ and tau-driven synaptotoxicity by preserving specific dendritic spine characteristics. (**A**) A separate cohort of 3xTg-AD female mice were injected using the schematic described and the brains removed immediately after the last injection and processed for Golgi–Cox impregnation as described in the methods section. (**B**) Representative dendrites from saline- and inhibitor-treated groups. The panel width corresponds to 10 μm. (**C**) The individual spine area (μm^2^) was significantly increased in the inhibitor-treated cohort compared to the saline-treated group (** *p* = 0.0048). (**D**) The perimeter of the selected dendrites was significantly greater in the inhibitor-treated cohort compared to the saline-treated group (** *p* = 0.0061). (**E**) The length of filamentous or thin spines per 10 μm was significantly greater in the inhibitor-treated cohort compared to the saline-treated group (** *p* = 0.0026). (**F**) Mushroom spine neck length (μm) was significantly increased in the inhibitor-treated cohort compared to the saline-treated group (**** *p* < 0.0001). (**G**) Mushroom spine neck width (μm) was significantly increased in the inhibitor-treated cohort compared to the saline-treated group (**** *p* < 0.0001). (**H**) Mushroom spine head length (μm) was significantly increased in the inhibitor-treated cohort compared to the saline-treated group (** *p* = 0.0074). (**I**) Mushroom spine head width (μm) was significantly increased in the inhibitor-treated cohort compared to the saline-treated group (* *p* = 0.0101). Each circle represents an individual spine. Saline circles have colored borders while inhibitor circles have black outline filled with color. For each animal, there were three slices assessed and for each slice there were five representative dendrites that were measured. Each panel has two graphs showing the same comparisons, the left shows the individual values in circles while the right shows the bar graph with the error bars. Values are expressed as mean ± SEM. n = 5 subjects per group (all females, 3 slices per animal and 5 representative dendrites per slice). Unpaired non-parametric Mann–Whitney test.

**Figure 5 ijms-24-03372-f005:**
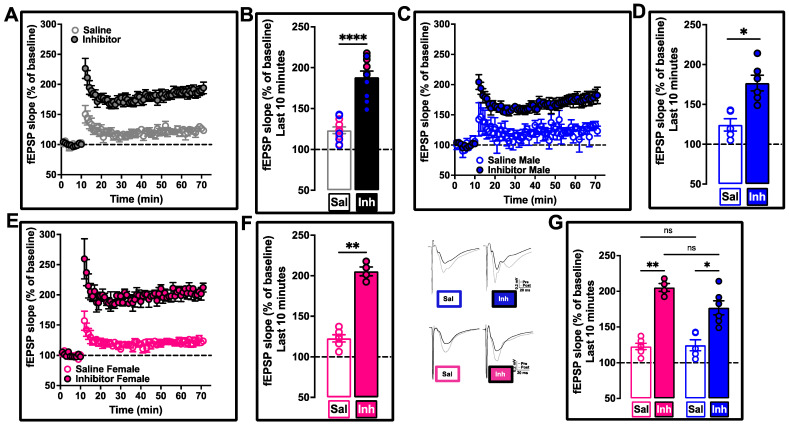
Preservation of HFS-LTP in the Schaffer collateral synapses by repeated VU01 treatment of 12-month-old 3xTg-AD mice. (**A**) Inhibitor-treated group (filled circles) showed increased potentiation compared to saline-treated sibling group (clear circles) following high frequency stimulation (HFS − 3 × 100 Hz). (**B**) The last 10 minutes (from 60–70 min in the previous panel) were averaged to obtain the mean and standard error of mean for each animal to plot the bar graph of saline-treated (clear bar) compared to PLD1 inhibitor-treated (filled bar). When the data were separated into males (**** *p* < 0.0001, n = 9–10 mice; unpaired non-parametric Mann–Whitney test). (**C**,**D**) * *p* < 0.0166; n = 5–6 mice, Uncorrected Dunn’s test) and females (**E**,**F**) ** *p* < 0.0015; n = 4–6 mice, Uncorrected Dunn’s test), the PLD1 inhibitor effect is seen in both sexes. We also assessed whether the sex-dependent increase showed differences between and within the treatment groups (**G**) ** *p* > 0.9999 for saline (male vs. female) and * *p* = 0.3491 for inhibitor (male vs. female); Uncorrected Dunn’s test, there was no significant difference, suggesting that the synaptic changes were comparable between the sexes. Representative traces for each sex (pink for females and blue for males) and treatment group (Sal vs. Inh) are provided. Each panel provides two overlapping sets of traces with pre-HFS (black) and post-HFS (grey) profile. Scale bars showing the amplitude in mV along the Y-axis (0.2 mV) and time in milliseconds (20 ms) along the X-axis for each panel. Each circle in (**B**,**F**,**D**,**G**) represents an animal. Saline circles have colored borders while inhibitor circles have black outline filled with color. Values are expressed as mean ± SEM. ns, no significance.

**Figure 6 ijms-24-03372-f006:**
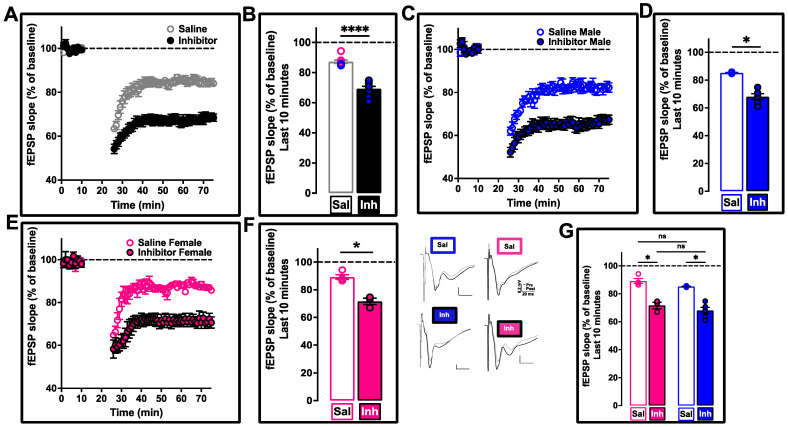
SC synapse LFS-LTD is preserved following PLD1 inhibition by VU01 in 12-month-old 3xTg-AD mice. (**A**) Inhibitor-treated group (filled circles) showed lower values of depression compared to saline-treated sibling group (clear circles) following low frequency stimulation (LFS − 900 X 1Hz − 15 min). (**B**) The last 10 minutes (from 60–70 min in the previous panel) were averaged to obtain the mean and standard error of mean for each animal to plot the bar graph of saline-treated (clear bar) compared to PLD1 inhibitor-treated (filled bar). The LTD in the inhibitor-treated group is significantly lower compared to the saline-treated group (**** *p* < 0.0001; n = 8–9 mice, unpaired non-parametric Mann–Whitney test). When the data were separated into males ((**C**,**D**) * *p* = 0.0318; n = 4–6 mice, Uncorrected Dunn’s test) and females ((**E**,**F**) * *p* = 0.0138; n = 3–4 mice, Uncorrected Dunn’s test), the differences were still present, validating that the PLD1 inhibitor effect is seen in both sexes. We also assessed whether the sex-dependent increase showed differences between and within the treatment groups ((**G**) * *p* = 0.2626 for saline (male vs. female) and * *p* = 0.6744 for inhibitor (male vs. female); Uncorrected Dunn’s test), there was no significant difference (ns), suggesting that the synaptic changes were comparable between the sexes. Representative traces for each sex (pink for females and blue for males) and treatment group (Sal vs. Inh) are provided. Each panel provides two overlapping sets of traces with pre-HFS (black) and post-HFS (grey) profile. Scale bars showing the amplitude in mV along the Y-axis (0.2 mV) and time in milliseconds (20 ms) along the X-axis for each panel. Each circle in (**B**,**F**,**D**,**G**) represents an animal. Saline circles have colored borders while inhibitor circles have black outline filled with color. Values are expressed as mean ± SEM.

**Figure 7 ijms-24-03372-f007:**
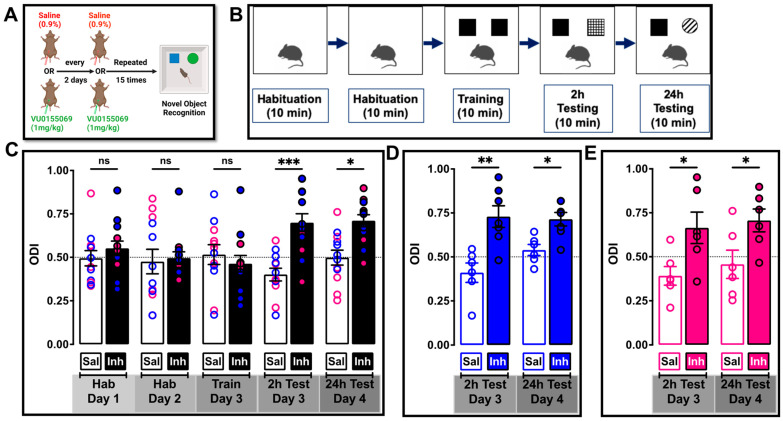
Chronic PLD1 inhibition rescues NOR deficits in 12-month-old 3xTg-AD mice. (**A**) Both male and female 3xTg-AD mice were injected (i.p.) with 1mg/kg VU01 or 0.9% saline every alternate day to receive a total of 15 injections for a period of 1 month. (**B**) The schematic depicts the regimen of the protocol described in the methods section. (**C**) The object discrimination index (ODI) measures the relative time spent by the animal in the novel object area compared to the total amount of time spent with both objects. There was no bias observed with any of the areas associated with either object in both groups of animals during habituation days (labeled in grey below X-axis, ns, not significant). On the training day, we did not observe any bias for the objects or the associated area for either group, suggesting that learning was not affected. There is a significant difference between the groups at the 2 h (*** *p* = 0.0003) and 24 h (* *p* = 0.0211) test where the inhibitor-treated group spent more time with the novel object compared to the saline-treated siblings. The saline-treated animals do not differentiate between the novel object and the familiar object relative to the PLD1 inhibitor-treated animals, providing evidence for compromised memory. Data were separated into (**D**) males (n = 6–7) 2 h (** *p* = 0.0014) and 24 h (* *p* = 0.0404) and (**E**) females (n = 6) 2 h (* *p* = 0.0275) and 24 h (* *p* = 0.0412) and PLD1 inhibition showed efficacy in both sexes. Saline-treated animals are shown in clear bars while inhibitor-treated animals are shown in filled bars. Each circle represents a single animal. Saline circles have colored borders while inhibitor circles have black outline filled with color. Pink color represents female, while blue represents male mice. Values are expressed as mean ± SEM, n = 12–13 mice, Holm–Sidak’s multiple comparison test and Kruskal–Wallis test.

**Figure 8 ijms-24-03372-f008:**
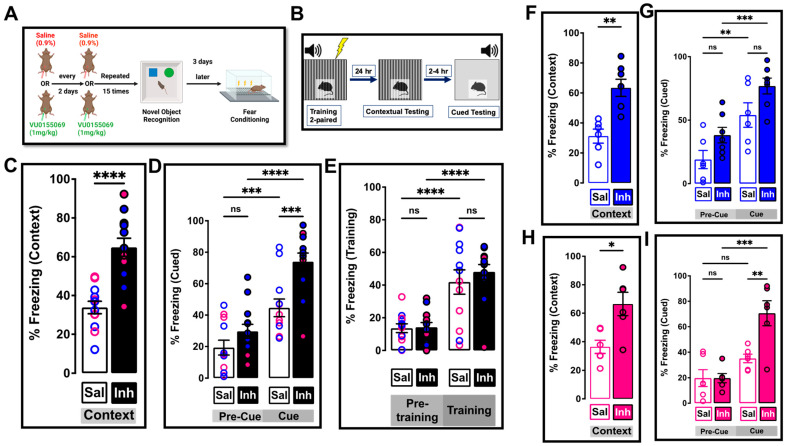
Chronic PLD1 injection prevents the cued and contextual memory deficit in 12-month-old 3xTg-AD mice. (**A**) The fear conditioned (FC) behavior was conducted in the same group of animals three days following the NOR test. (**B**) Schematic for FC training and testing is described in detail in the methods section. After a two-paired training regimen, animals were returned to their cages for 24 h prior to testing in the same context. After contextual testing, the animals were returned to their cages. After 2–4 h, the animals were introduced to a new chamber with different visual, tactile and olfactory characteristics. After a brief pre-cue period where the baseline level of freezing was assessed, the animals were given the sound cue. Once the behaviors were completed, the animals were returned to their cages. (**C**) During the contextual phase, the saline-treated animals show a greater level of activity (less % freezing) compared to the inhibitor-treated group, suggesting that more of the inhibitor-treated animals had retained the contextual memory associated with the shock (**** *p* < 0.0001). (**D**) In the freezing test, the pre-cue (before the sound is played) freezing response is not different (ns) between the two groups of 3xTg-AD mice. The post-cue response shows a significantly increased freezing response for saline and inhibitor compared to their respective pre-cue responses (*** *p* = 0.0005). Contrary to what we reported in 6-month-old mice, we report observing a significant difference in the cued response between the VU01-treated and saline-treated group, suggesting that the increased amyloidogenic insults show compromised cued memory at 12 months of age in this mouse model. (**E**) There were no differences in the two groups of animals in the extent of freezing before and during training. The increased freezing response for both the saline- and the inhibitor-treated groups was significant compared to their respective pre-shock response. Data were separated into (**F**) or (**G**) males and (**H**) or (**I**) females, where saline-treated animals are shown in clear bars while inhibitor-treated animals are shown in filled bars. In the contextual response, that mirrors the NOR response, both (**F**) male (** *p* = 0.0012; n = 6–7 mice) and (**H**) female (* *p* = 0.0152; n = 6 mice) show improvement in their contextual memory compared to their saline-treated siblings. Interestingly, in the cued response, (**G**) the post-cue freezing in the saline-treated male (clear bars and clear circles) mice does not reach significance compared to pre-cue (Sal: 54.010 ± 9.649 vs. Inh: 76.900 ± 6.201; *p* = 0.0571; n = 6–7 mice), but (**I**) the difference in response between the female treatment group does show significance (** *p* = 0.0036; n = 6 mice). When examining the individual responses, the variability in the male response, post-cue, is more compared to the females. More importantly, the inhibitor-treated animals (filled bars and filled circles) in both male (*** *p* = 0.0003) and female mice (**** *p* = 0.0001) are significantly different compared to pre-cue and show no difference between the sexes, suggesting that the treatment with the PLD1 inhibitor has similar effects between the sexes and promotes protection against the Aβ and tau-driven insults on cued (amygdala-dependent) memory expression. Saline-treated animals are shown in clear bars while inhibitor-treated animals are shown in filled bars. Each circle represents a single animal. Saline circles have colored borders while inhibitor circles have black outline filled with color. Pink color represents female, while blue represents male mice. Values are expressed as mean ± SEM, n = 12–13 mice, unpaired non-parametric Mann–Whitney test for contextual and uncorrected Fisher’s LSD for cued.

**Table 1 ijms-24-03372-t001:** Deidentified human clinical samples used in the study are represented with respect to the data that were made available post-mortem with their case numbers, diagnosis as non-pathological (CTRL) or pathological (AD) for the presence of plaques and tangles, age, gender, Braak staging, (Mini Mental State Examination) MMSE scores and post-mortem interval (PMI) in hours.

Case No.	Diagnosis	Age (Years)	Gender	Braak Stage	MMSE	PMI (h)
1220	CTRL	>89	F	2	30	12
2467	CTRL	99	F	3	28	4.5
2553	CTRL	100	M	2	28	4
2682	CTRL	90	F	2	29	9
2755	CTRL	95	F	2	29	18
2953	CTRL	100	F	3	27	2.5
1678	AD	76	F	6	1	25
2312	AD	87	F	6	-	2.5
2316	AD	83	M	5	-	13
2317	AD	88	M	6	-	4
2374	AD	91	M	6	-	24
2543	AD	95	M	6	21	5

## Data Availability

All data generated or analyzed during this study are included in this published article [and its Appendix A].

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
