# Peer review of "Phospholipase D1 Attenuation Therapeutics Promotes Resilience against Synaptotoxicity in 12-Month-Old 3xTg-AD Mouse Model of Progressive Neurodegeneration"

_ijms, 2023, doi:10.3390/ijms24043372_

Round 1

Reviewer 1 Report

Thank you for your hard work. The reviewer can clearly see a lot of experimental work was involved. I have some minor comments:

1. Would you plan to investigate various ages of the mice model as well? Would you expect the age of the mice to have an impact?

2. Would the concentration of VU01 have an impact? Have you studied other concentrations?

3. Would the injection interval of VU01 have an impact?

Author Response

We thank the reviewer for their time, effort and consideration. We appreciate the excellent input that would allow us to refine the future studies accordingly. With respect to the questions posed by the reviewer - please find the following responses:

  1. Would you plan to investigate various ages of the mice model as well? Would you expect the age of the mice to have an impact?
    • According to several published studies by different groups, including the original set of publications by the LaFerla group that generated the 3xTg-AD mice, the significance of the mice is that there is progressive accumulation of Aß and tau over time and as a result, there is an increase in the cognitive decline that mirrors to some extent the progression in human population based of these amyloidogenic proteins. As a result, we are pursuing studies at this point on 18-month old mice that have severely compromised cognition as well as many physical disabilities as well as sex-specific changes noted in literature with an intent of uncovering whether there are some influences associated with aging. Once we complete these set of studies, we are planning to conduct proteomic and transcriptomic analysis that would facilitate a better understanding of the therapeutic effects.

2. Would the concentration of VU01 have an impact? Have you studied other concentrations?

  • We strongly suspect that the concentration would have an effect and we kept it to the lowest possible dose based on our previous study because the intent is attenuation and not elimination which could give rise to other homeostatic processes that may confound the therapeutic outcome. Nevertheless, we are in the process of refining the PK/PD and ADME studies so that we can get a complete understanding of the tolerability of VU01.

3. Would the injection interval of VU01 have an impact?

  • We would suspect that this would be definitely a possibility to explore as we advance further towards therapeutics, we are also planning these with different vertebrate models to check on the possibility of different dosages and routes to ensure that the optimal dose is available. These are excellent suggestions that we hope to pursue and publish in the future towards making it possible to treat many of the cognitive decline associated with Alzheimer's disease and related dementia.

Reviewer 2 Report

In this work, the authors show that PLD1 increases in the hippocampus of AD patients and different hippocampus regions of the 12-month-old-3xTg-AD mice, and PLD1 signal colocalizes with the signal for the amyloid beta peptide and t Tau protein aggregates. The work also describes that 11-month-old 3xTg-AD mice IP treated with VU0 155069 (VU01) inhibitor for one month show better novel object recognition (NOR) and fear conditioning (FC) cognitive performance than saline-treated mice. In addition, 3xTg-AD animals treated with VU01 show moderate increases in the area, size, and length of the mushroom-like dendritic spines. The hippocampi of these 3XTg-AD mice presented LTP and LTD of greater magnitude when stimulated with high or low frequency, respectively, than the 3xTg-AD mice treated with saline.

The work presents many good-quality results and is well-written. The introduction antecedents are adequate, although previous papers that describe PLD1 alterations in AD are not included, nor are the papers showing that PLD1 modulates APP processing. 

The methods, results, the numbers of animals, and statistical analyses are well described, in general, it is a job done competently. Unfortunately, the novelty of the work is very limited. In previous works, the group already had shown increased expression of PLD1 in samples from AD patients. In Krishnan et al., 2018 (Elevated phospholipase D isoform 1 in Alzheimer's disease patients' hippocampus: Relevance to synaptic dysfunction and memory deficits. Alzheimer's Dement), authors showed that the PLD1 levels are increased in synaptosomal fractions from hippocampus/temporal cortex of AD patients as well as in brain of 3xTg-AD mice. Also, other works have described increases in Phospholipase D1 in Alzheimer's disease (Jin et al., 2006, Jin et al., 2007) and its association with amyloid precursor protein processing modulation.

Neither the novelty of the treatment of 3xTg-AD mice with VU01 is clear. Previously the authors reported that 3xTg-AD mice treated with VU01 showed better performance in novel object recognition (NOR) and fear conditioning (FC) tests than saline-treated mice (Bourne et al., 2019 (Suppressing aberrant phospholipase D1 signaling in 3xTg Alzheimer's disease mouse model promotes synaptic resilience Scientific Reports). Moreover, the results reported here are very similar to the previous ones. The same happens for the increases in LTP, dendritic spine density, and mushroom spine numbers and size.

The authors argue that the age of 3xTg-AD mice used in both studies is different, 6 months in the previous work and 12 months in this study, the increases in cognitive test performance by the VU01 are similar. However, it is not possible to know if the VU01 effects are larger or smaller or establish some relationships between beta-amyloid pathology and Tau pathology progression contrasting the cognitive and synaptic function results of both works, moreover, because the aged-matched WT mice are not included.  Then, the conceptual advance of PLD1's role and the usefulness of VU01 to prevent AD pathology progression it is not so clear, these already have been reported.

Moreover, in the paper, Krishnan et al., 2018 the authors reported that the VU01 inhibitor prevents the decrease in LTP of 3xTg-AD mice in 6- and 18-months old mice, and these results are compared to wild-type mice of the same ages. Thus, it has already been described that VU01 prevents synaptic dysfunction in the early and late stages of 3xTg-AD mouse pathology, and this correlated with improvements in memory and learning. Thus, the main weakness of the work is that it is not new and does not provide new relevant information on the role of PLD1 in the progress of AD pathology. Another weakness is that the intraperitoneal treatment with not translational, it could be discussed if it passes the BBB or if there are other PLD1 inhibitors with better brain penetrance.

In general, the paper is well written, and statements and conclusions are supported by the data. However, in some parts, especially in the figure legends, some sentences over concluded.  As follows I will review the results statements and conclusions and if the results support these.

2.1. Increased PLD1 levels and increased co-localization with amyloidogenic proteins in postmortem human AD hippocampal slices compared to age-matched control.

OK. The results support this statement. The work shows the increase in the PLD1 signal in the hippocampi of AD patients and 3xTg-AD mice. Also, the data shows that the PLD1 signal co-localizes with the Ab and Tau aggregates. However, it does not necessarily indicate an interaction between these proteins the beta-amyloid and tau aggregates are sticky, and many proteins get included in these aggregates in the AD brain.

It is necessary that the papers that report APP and PLD1 association to be mentioned in the discussion. 

2.2. Chronic one-month treatment with PLD1 inhibitor (VU0155069 or VU01 at 1 mg/kg/2 days i.p.) in 12-month-old-3xTg-AD mice attenuates PLD1, PLD1-Aβ and PLD1-Tau co-staining expression differentially in hippocampal subregions.

The results partially support this statement. Although immunofluorescence is not a quantitative method, increases in the PLD1 signal and the co-staining for Aβ and Tau were observed. However, the mechanism by which the inhibitor would reduce the expression of PLD1 is not clear and is not explored in the work.

The potential mechanism that upregulated PLD1 in AD is very little developed in the discussion. Only potential feedback between Aβ or Tau and synapse protective effect is discussed.

23. VU01 dependent chronic inhibition of PLD1 preserves specific dendritic spine morphologies in 12-month-old 3xTg-AD mice.

This affirmation is only partially supported by the data. It was found that the 3xTg-Ad mice treated with VU01 present mushroom spines slightly greater in area, perimeter head size, and length of the neck than the saline. However, the dendritic spines of aged-matched WT mice were not analyzed, then it is not possible to affirm that the spine morphologies are preserved without comparison with WT. 

The changes are modest, and controls of WT animals are required.

2.4. Restoration of hippocampal synaptic function by chronic PLD1 inhibition regimen in 12-month-old 3xTg-AD mice.

Again, the data support the affirmation, but it is necessary data from age-matched WT mice to be able to argue that inhibitor treatment restores synaptic function. The 3xTg-AD Inhibitor-treated group showed potentiation with values of fEPSP slope around 180 compared to a saline-treated sibling group of 120. LTP values of 3xTg-AD mice treated VU01 approached to aged-matched WT group reported values  (fEPSP slope around 220) paper Krishnan et al., 2018. However, is necessary to include the aged-matched WT group data to support the affirmation that PLD1 inhibitor restores synaptic function.

Regarding the LTD, although again the inhibitor-treated group showed lower values of LTD compared to a saline-treated sibling group, the values of the WT are not presented.

2.5. Chronic one-month treatment with PLD1 inhibitor ameliorates memory deficits in 12-month-old 3xTg-AD mice.

OK, the results support this claim. The animals treated with VU01 presented a better performance in the NOR and FC tests. However, as mentioned above to support and compare the capacity of VU01 to reduce the cognitive deficits of the 3xTg-AD mice at different ages is necessary to have age-matched WT data. Although the authors are careful in the description of the results, sometimes they say VU01 rescue or prevent, and this cannot be determined without the values of the aged-matched WT animals.

Although comparing the effects of VU01 in 12 months mice, when both amyloid and tau pathologies are observed, with those observed in 5 months mice, when supposed would only be Ab pathology, would give novelty to the work, the fact that the groups correspond to different papers and that the WT groups with the respective ages are not available to allow comparison, comment on possible age-dependent effects does not seem solid. 

Some studies were carried out and analyzed as males and females separated groups. However, no relevant differences were observed.

Minor points:

In Figure 2, the photos of hippocampus regions are in different orientations, this can be improved for better comparison.

Figures 5A and 6A the gray-filled circles are difficult of distinguishing from the open circles.

Figures 4H and 4I the legends of the graph are missing 

Author Response

We would like to take this opportunity and thank the reviewer for conducting a very thorough and very constructively critical analysis of the manuscript. We are attempting to provide an explanation as well as the corrective actions that we have taken for all the comments that the reviewer has so graciously provided us. 

We thank the reviewer for the compliments on the writing of the manuscript and the appreciation for the body of work and effort that went in from the team.

With regards to the need for novelty, we would like to apologize for our lack of clarity. The work is novel in many regards that we explain here and have included in the body of the amended manuscript (highlighted in brown color and underlined) for the reviewer's perusal. We have now included an introductory paragraph that highlights the previous work done on PLD and how Jin et al. literature that showed elevated PLD levels and reiterated the fact that 3xTg-AD mice provide us with the unique opportunity to study what happens with Abeta accumulation (that occurs from 3-months of age) and the importance of 12-month-old mice and looking at tau-related synaptic dysfunction and the effect of inhibition that occurs after the full-blown effects occur. We have emphasized that the study is looking at the prospect of whether PLD1 inhibition done for a limited time mimicking what would happen if the drug was given after the clinical symptoms were completely occurring (like in a 11-month old mouse with full blown Abeta and Tau driven cognitive decline), whether our intervention would still be useful - this is an important followup to our previous work that merits reporting for the scientific audience. 

More importantly, this study is among the few to address whether therapeutic intervention with the same dosage of VU01 after full-blown effects of Ab and tau is efficacious, a premise that would extremely important when advancing to clinical trials to estimate whether this intervention can be applied at early stages or even when it is late-stage.  

We apologize for not being clear enough, but we have attempted to use the same dosage because it is important to suppress but not completely eliminate PLD1, this alone will not change the homeostasis and therefore show more efficacy as is well established by the pharmacological basis of tolerance to drugs. As a result, we have used the same dosage but with attention to more details of the experiments, including epochs to see whether there are early changes that are missed, which were not apparent here, but may manifest in 18-month-old, where there is increased deposition, accumulation that is known to affect the cognitive status even more in terms of the speed with which the cognitive activity occurs. Therefore, we humbly propose that such a thorough and systematic investigation of every drug that is being administered in neurodegenerative states needs to be reported so that there is enough consensus from the esteemed scientific community to support future studies that would lead to therapeutic application and human pilot clinical trials with confidence. Moreover, we do show, even in our behavior, that compared to the 6-month-old mice, there are sex-specific differences that are observed in fear conditioning thereby increasing the importance of reporting such changes. We have now emphasized the point that the change that we saw was not with the drug-treated group but with the untreated group, highlighting why it is important to conduct this study not with wild type mice, but with mice having the same background as the experimental mice from a therapeutic point of view. We are conducting studies with WT mice where we are using different mechanistic approaches to understand the mechanism, but that is beyond the scope of this work that is dedicated to therapeutic advances - but we appreciate that excellent point from the reviewer - that exhaustive body of work is currently in progress and will be the subject of our reporting in the future.

We would like to emphasize that there is LTD studies that were also conducted here. LTD is a measure that is standard for tau-related pathology and this is a novel proof that we have included here in this study and the only one in the literature that has been shown in 3xTg-AD mice that we are aware of. This novel finding shows that there is importance in looking at the effects of the PLD inhibitor affecting the mechanism of glutamate neurotransmission - a complete regulation that we are focusing on.

The reviewer asks a very relevant question regarding the ability of the drug to cross the blood-brain barrier - we have demonstrated that the drug is efficacious using multimodal functional assessments and compared it to age-matched siblings of the same genetic background - which is the strength of the study. We have enhanced the importance and the relevance by showing the PLD1 levels are indeed decreased in the inhibitor treated group in the hippocampi, thus establishing that the drug crosses the blood brain barrier. Moreover, as the reviewer points out there are effects that are observed in the reduction of Abeta  with VU01 administration as noted in Fig. 2 and that is also evidence for the drug crossing the blood brain barrier.

We agree with the reviewer on his observations about Fig. 1 and we have included a statement that highlights that there is a role for PLD1 with APP that is highlighted by our results and included a sentence that refers to the PLD1 association with APP from earlier studies.

We have now included statements speculating putative mechanisms that may be important in VU01 function as per the reviewers recommendation.

With respect to looking at dendritic spines with attention to the WT profile, this is something that is beyond the scope of this study because here we are exploring the therapeutic potential in cessation of progression and not necessarily improvement to the WT like states, besides the improvement being modest is representative of the drug efficacy and we have included this statement in our discussion section to accommodate the reviewers astute analysis.

In order to address male and female differences more elaborately, greater power would be needed, but we did parse the samples separately since AD affects females disproportionately and we believe that the therapeutic value should therefore take into consideration the sex of the test subject for any functional analysis. We do report a change in behavior, but not in the VU01 treated, but the saline treated 3xTg-AD which again highlights the importance of why this therapeutic study needs 3xTg-AD as a control compared to WT control that would introduce more variables that cannot be adequately accounted for or confidently addressed by the rationale associated with the parameters of our study.

The stack image planes of the figures align with the orientation of the hippocampi and therefore, we chose to provide the images as we capture them to avoid any kind of misrepresentation or fudging that may confound the integrity of the data for Fig. 1-3.

Figure 5A and Figure 6A have now been increased and the circles are now either clear or filled for better contrast

Figure 4H and Figure 4I - the legends are provided.

We have also increased the figure size for visibility to address the right concerns of the reviewer.

Round 2

Reviewer 2 Report

The changes made in the paper text, especially in the introduction and discussion, have improved the paper. However, the improvement is not enough.  This new paper version presents a better highlighting of which is a novelty with respect to previous publications of the group. However, the contribution to knowledge about the role of PLD1 and the clinical potential of VU01 for the treatment of Alzheimer's disease is not a substantial increase.

Thus, the main paper's weakness is still the limited or low novelty. Maybe authors would have evaluated the mechanisms by which PLD1 increases in 3XTg-AD mice or how PLD1 inhibition modulates neurodegeneration, they could have increased the originality and impact of the work. In any case, the work is well done, with clear and well-described results. Moreover, the introduction and discussion improved in this revised version.

Furthermore, the paper still presents problems with the clarity of the graphs in figures 5 and 6; the filled and clear circles are not distinguishable. In addition, it would be convenient to use clear lines without blurred contours. In figures 2 and 3, the hippocampus regions images are not in the same orientation, these previous observations were not improved either.

Author Response

We thank the reviewer for his acknowledgement of our effort to improve the introduction and discussion, whose course was immensely helped with the insights and constructive criticism provided by the reviewer.

With respect to the limited novelty that the reviewer has reiterated, we have now expanded the discussion section to show how the findings that the reviewer directed our attention to, namely astroglial aberrant elevation of PLD1, may be a key player in driving late onset Alzheimer's disease (LOAD). This increases the novelty of our finding that could not be demonstrated without highlighting systematically the therapeutic efficacy at 12-months, keeping every other aspect constant, as we have elaborated in the discussion section. We hope that this would address the concerns of the reviewer in terms of novelty. Additionally, we are commencing some collaborative studies with MODEL-AD group as well as our Mitchell Center group to perform some deep proteomic and transcriptomic studies that will be functionally evaluated along the speculations that we have made in the discussion - we thank the critical constructive and insistent insight that allowed us to think out of the box - we value this greatly.

As the reviewer suggested, we have changed the orientation of the figures 2 & 3 to match each other and used different symbols to enhance the visibility of figures 5 & 6. Also, all the shadowing for all figures have been removed to improve the contrast. We thank the reviewer for this excellent and required constructive criticism.